# SMART: Self-Weighted Multimodal Fusion for Diagnostics of Neurodegenerative Disorders

***

***

***

## ABSTRACT

Multimodal medical data, such as brain scans and non-imaging clinical records like demographics and neuropsychology examinations, play an important role in diagnosing neurodegenerative disorders, e.g., Alzheimer's disease (AD) and Parkinson's disease (PD). However, the disease-relevant information is overwhelmed by the high-dimensional image scans and the massive non-imaging data, making it a challenging task to fuse multimodal medical inputs efficiently. Recent multimodal learning methods adopt deep encoders to extract features and simple concatenation or alignment techniques for feature fusion, which suffer the representation degeneration issue due to the vast irrelevant information. To address this challenge, we propose a deep self-weighted multimodal relevance weighting approach, which leverages clustering-based constrastive learning and eliminates the intra- and inter-modal irrelevancy. The learned relevance score is integrated as a gate with a multimodal attention transformer to provide an improved fusion for the final diagnosis. Our proposed model, called SMART (Self-weighted Multimodal Attention-and-Relevance gated Transformer), is extensively evaluated on three public AD/PD datasets and achieves state-of-the-art (SOTA) performance in the diagnostics of neurodegenerative disorders. Our source code will be available.

## KEYWORDS

Multimodal Learning, Self-Supervised Clustering, Contrastive Learning, Neurodegenerative Disorder Diagnostic.

**ACM Reference Format:**
***. 2024. SMART: Self-Weighted Multimodal Fusion for Diagnostics of Neurodegenerative Disorders . In *Proceedings of Make sure to enter the correct conference title from your rights confirmation emai (Conference acronym 'XX)*. ACM, New York, NY, USA, 10 pages. https://doi.org/XXXXXXX.XXXXXXX

## 1 INTRODUCTION

Neurodegenerative disorders [19], e.g., Alzheimer's disease (AD) [9, 49], Parkinson's disease (PD) [8, 26], affect millions of people worldwide; unfortunately, they are currently incurable. However, early diagnosis of neurodegenerative diseases is crucial for intervention and provides people affected by dementia the opportunity to access early treatments and make plans for future care. In the clinical

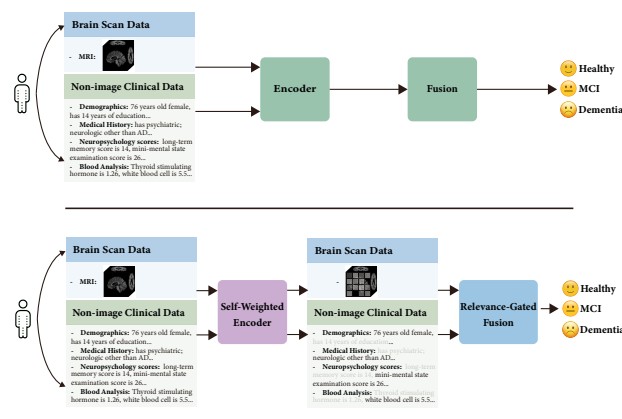

**Figure 1: Conceptual comparison between prior state-of-the-art methods (top) [5, 60, 64] and our approach (bottom) for the multimodal diagnostics of neurodegenerative disorders.**

diagnosis of neurodegenerative disorders, medical data commonly exhibits a multi-modal nature, including medical image scans like brain MRIs and non-imaging clinical information, e.g., demographics, serological tests, neuropsychology examinations, etc. The abundance of high-dimensional medical scans and vast clinical data overwhelms the disease-related information. Therefore, how to distinguish and fuse task-discriminative information among these multimodal clinical data is essential but challenging for an accurate computer-aided diagnosis of neurodegenerative disorders.

In the early stage, machine learning-based methods require manual selection of disease-related features for diagnosis [40, 47, 56]; however, the related expert knowledge is too limited to support an accurate selection of diagnostic features. Along with the popularity of deep learning (DL) approaches, automatic feature selection has become a top choice, and most existing methods completely depend on DL techniques to handle the feature selection and fusion task among massive information. As illustrated in Fig. 1, a typical approach adopted by most researchers [15, 28, 64] is extracting features for imaging and non-imaging data separately and then concatenating them for diagnostic classification. Since task-related and irrelevant information is intermingled, it is exhausting and overwhelming to fuse so much information for diagnosis. Also, due to the space discrepancy of multimodal features and the demonstrated effectiveness of recent vision language pre-training (VLP) models, e.g., CLIP [45], ALBEF [32], BLIP [31], a couple of recent works, e.g., MedCLIP [55], Alifuse [5], align imaging features to the text feature space of the non-imaging data and fuse them using a pre-trained large language model or the cross-attention techniques.

However, VLP-based models prefer paired image and text data, such as medical scans and reports [58], where texts describe the associated medical images. Differently, the non-imaging data for clinical diagnosis is partially paired with medical scans and provides complementary diagnostic information about a subject. More importantly, the alignment between imaging and non-imaging happens at the image level, it lacks the fine-grained features that help fuse partially-paired data and contribute to the disease diagnosis.

Considering the existence of a large amount of irrelevant information for disease diagnosis in both imaging and non-imaging data, in this paper, we explore the possibility of automatically highlighting and fusing the multimodal features that have a strong correlation with the target disease, as shown at the bottom of Fig. 1. The disease-irrelevant information exists at both intra-modal and inter-modal levels. To eliminate this irrelevancy issue, we propose a Self-supervised Multimodal Relevance Weighting (SMRW) module, which hierarchically clusters relevant information and learns a relevance score vector to weight within and among multimodal data. Guided by the learned weights, the multimodal feature fusion becomes relatively easy via a transformer with attention mechanisms. Hence, we propose a Multimodal Relevance-gated Attention Transformer (MRAT), which involves intra-modal and inter-modal relevance scores in calculating self-attention and image-to-text and text-to-image cross-attention, resulting in an efficient relevance-gated fusion for the diagnostic prediction. Figure 2 depicts the overview architecture of our proposed model, i.e., Self-weighted Multimodal Attention-and-Relevance gated Transformer (SMART), which is applied to the diagnostics of neurodegenerative disorders.

Overall, the contributions of this paper are summarized below:

(1) We propose a novel framework SMART for multimodal neurodegenerative disorder diagnosis. Extensive experiments on three public benchmark datasets for neurodegenerative disorders like AD and PD demonstrate the superiority of our approach over ten baselines, including previous SOTA methods.

(2) We propose a self-weighted multimodal representation learning technique SMRW, which adopts a self-supervised two-level contrastive learning to automatically cluster and weight relevant information at both intra-modal and inter-modal levels. A follow-up relevancy-gated attention module allows an efficient multimodal feature fusion for the final prediction.

(3) Thanks to the relevancy score learned by SMRW, our model is explainable to some extent while having a high diagnostic accuracy. Also, our model is theoretically designed for multiple modalities, which could include more modalities like audio to fully leverage all possible medical information.

## 2 RELATED WORK

### 2.1 Vision-Language Model (VLM)

Vision-language representation learning aims to jointly encode vision and language in a fusion model, which has been demonstrated to learn uni-modal and multi-modal representations with superior performance on downstream tasks [32, 45]. CLIP [45] and ALIGN [24] are dual-encoder models that are pre-trained with contrastive learning objectives on image-text pairs. They learn strong image and text unimodal representations with simple multimodal

alignment, which is not enough to handle tasks that require complex reasoning. ALBEF [32], CoCa [61], and BLIP [31] promote a deeper interaction between images and text using a deep fusion encoder with cross-modal attention. These models achieve better performance for vision-language classification by learning efficient multi-modal representations. Medical image-text representation learning is investigated based on contrastive learning as well, such as MedCLIP [55], CheXZero [52], and MedViLL [38]. However, these methods prefer modeling on paired image and text data, such as medical scans and reports.

### 2.2 Multimodal Learning in Medical Diagnosis

In medical diagnosis, fully leveraging medical data collected from multiple modalities becomes a popular choice, since multimodal machine learning models can leverage more information and more easily identify patterns of diseases, compared to using a single modality [2, 7, 12, 35, 51, 59]. A straightforward way for multimodal learning is to concatenate features and feed them into a classifier such as SVM or MLP for prediction [14, 21, 40, 43, 48]. To improve the model performance, feature selection methods are applied to reduce the feature dimension [18, 47, 63, 65]. Compared to machine learning methods, deep learning methods are feasible to capture hierarchical representations and achieve better performances. Kim et al. [28] propose a heterogeneous graph learning method to fuse the multimodal medical data. Zhou et al. [64] introduces the IRENE model based on Transformers, which fused representations among modality-specific low-level embeddings for diagnosis. Also, multimodal data like images and genomes have been used in diagnostics of breast cancer [15]. However, most of these studies potentially ignore the heterogeneity between modalities due to a lack of ability to fully explore the task relevance in intra-and-inter modalities, resulting in sub-optimal results.

### 2.3 Contrastive Learning

Contrastive learning [17, 41, 44] pulls positive pairs closer and pushes negative pairs farther contrastively to obtain discriminative features. In addition to the single-modality representation learning, contrastive methods for multiple modalities are also widely explored. Multi-modality data often contain two or more modalities and they naturally form multiple inputs. The common methods [3, 24, 27, 32, 45, 50, 62] leverage the cross-modal contrastive matching to align two different modalities and learn the inter-modality correspondence. However, since the semantic consistency among intra-modal and inter-modal is not guaranteed, it is challenging to capture the useful information in multimodal data, while considering the side effects of irrelevant information. For example, Yang et al. [60] leverages the maximization of mutual information to conduct consistency learning across different views and aims to achieve a provable sufficient representation. Xu et al. [57] observe the conflict between consistency and reconstruction objectives in encoder-decoder frameworks and propose to learn multi-level features for multiple modalities. Li et al. [34] propose leveraging domain-specific medical knowledge as guiding signals to perform multi-level contrastive learning. Although satisfactory results are

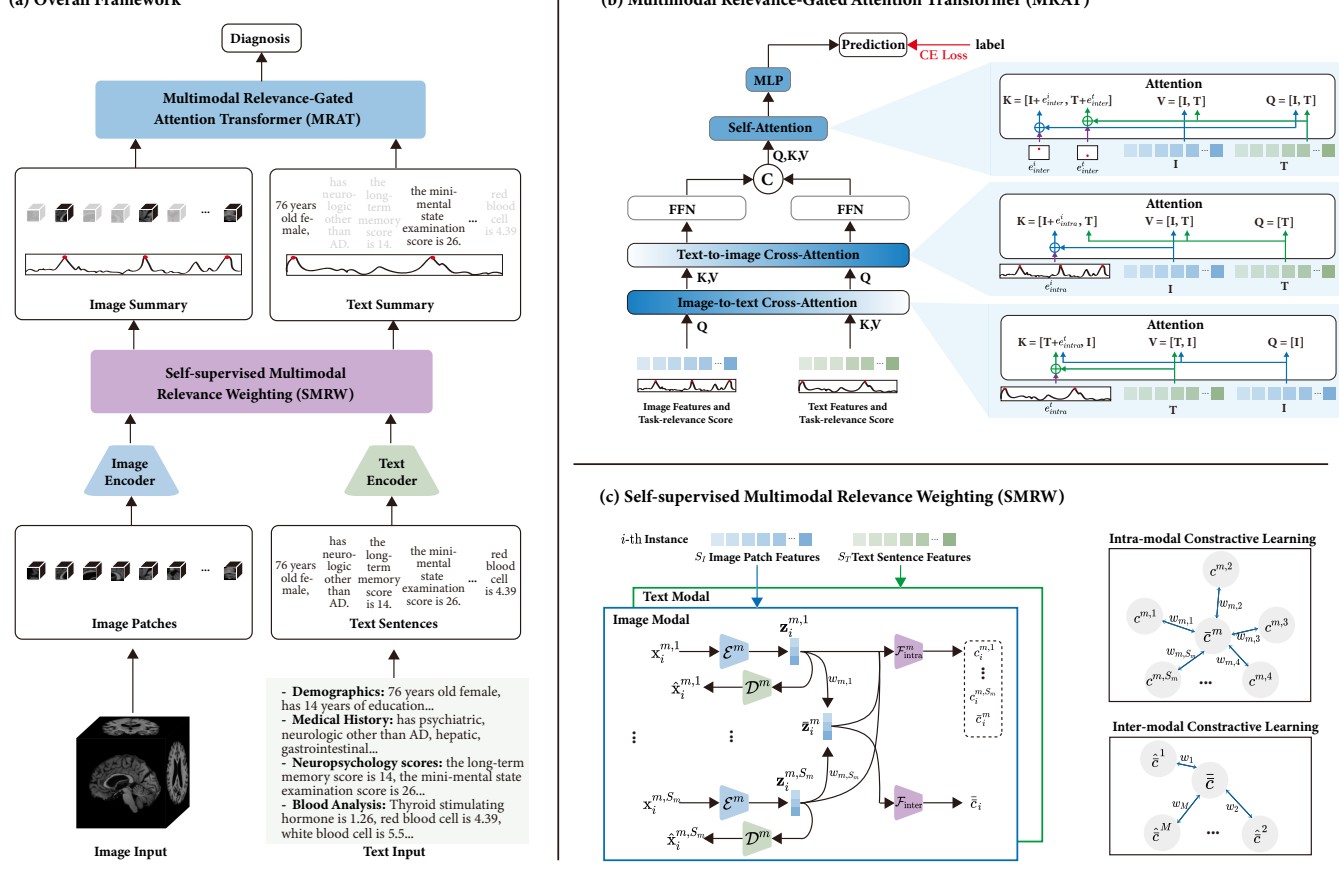

**Figure 2: (a) An overview of our proposed SMART network for neurodegenerative disorder diagnosis, where two visual and textual branches are jointly trained. The output of the fusion branch predicts the diagnostic outcome. (b) Our Multimodal Relevance-Gated Attention Transformer (MRAT) takes the features and task-relevance distributions from two modalities as input and generates disease probability. (c) A detailed diagram of Self-supervised Multimodal Relevance Weighting (SMRW) works in a self-weighted mode to summarize task-relevant information within and between image and text modalities.**

achieved in many cases, the multimodal representation degeneration is still not well considered and addressed. Recently, new perspectives of deep clustering have been offered by the development of contrastive clustering [33]. In this paper, we propose a cluster-based self-weighted intra-modal and inter-modal contrastive learning to address the irrelevancy issue in multimodal medical diagnosis.

## 3 METHODOLOGY

Figure 2 presents the architecture framework of our SMART network. The input to our model is the multi-modality (e.g., brain structure MRI and clinical transcribed text in our case). Our model includes three components: the feature encoding, the Self-supervised Multimodal Relevance Weighting (SMRW) module, and the Multimodal Relevance-gated Attention Transformer (MRAT) module.

### 3.1 Feature Encoding

We extract unimodal features for brain structure MRI and text-described clinical data with independent encoders to obtain their

hidden representations. Following [64], we use ViT [11] as the feature encoder for each patch of brain structure MRIs and BERT [10] as the feature encoder for each sentence of the textual description of non-imaging clinical data. Specifically, we denote the generated image and text features as $\mathbf{I} \in \mathbb{R}^{S_I \times D_I}$ and $\mathbf{T} \in \mathbb{R}^{S_T \times D_T}$, where $S_I$ indicates the number of patches of a medical image and $S_T$ indicates the number of sentences of the clinical text, respectively. Features of different modalities are then projected into a common embedding space with the same dimension using a linear fully connected layer.

### 3.2 Self-supervised Multimodal Relevance Weighting (SMRW)

In multi-modality learning, variations in data quality within and across modalities are common, posing a risk of representation degradation when merging data or features. Specifically, this degeneration makes the representation of task-relevant data mediocre, potentially leading to the loss of crucial discriminative information, as observed in Fig. 3. We take a multi-modal dataset ADNI [42] as an example. We first evaluate the representation quality of containing

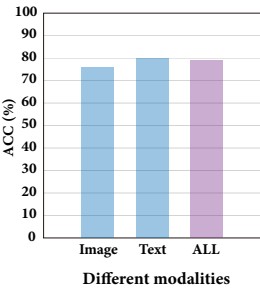

**Figure 3: Diagnostic accuracy on the ADNI dataset using various data types within a model and across different modalities.**

semantics of different categories of clinical information except for image scans, including SC (Subject Characteristics), MH (Medical History), NS (Neuropsychology Scores), BA (Blood Analysis), GN (Genetic), and All of them. Concatenating all categories of clinical semantic information performs worse than simply using NS in AD diagnosis. Similar phenomenon is observed in learning multimodal representation using images and text.

To mitigate such representation degeneration issues in multimodal learning, we seek a technique that upgrades the task-relevant information while downgrading the task-irrelevant one, so that all intra-modal and inter-modal relevant information can be highlighted for further fusion. Hence, as shown in Fig. 2(c) we propose a Self-supervised Multimodal Relevance Weighting (SMRW) module based on cluster contrastive learning, which learns distinct representations for intra- and inter-modal relevant information.

*3.2.1 Intra-modal Constrastive Learning.* Consider an input $\mathbf{X} = \{\mathbf{X}^m \in \mathbb{R}^{N \times S_m \times D_m}\}_{m=1}^M$ with $M$ modalities (e.g. $M = 2$, image and text modalities in our case) and N instances (e.g. the number of patients in our case); $S_m$ denotes the total items (e.g. image patches or text sentences) of the $m$-th modality, $D_m$ denotes the feature dimension of the $m$-th modality. $\mathbf{X}^{m,s} = [\mathbf{x}_1^{m,s}, \mathbf{x}_2^{m,s}, ..., \mathbf{x}_N^{m,s}]$ denotes the instance collection of $\mathbf{x}_i^{m,s}$ from the $m$-th modality's $s$-th sub-items of the $i$-th subject. We designate the encoder $\mathcal{E}^m$ and the decoder $\mathcal{D}^m$ for the specific $m$-th modality. Encoder $\mathcal{E}^m$ projects the raw data into modality-specific feature space via

$$\mathbf{z}_i^{m,s} = \mathcal{E}^m \left( \mathbf{x}_i^{m,s} \right), \tag{1}$$

resulting in $\mathbf{Z}^{m,s} = [\mathbf{z}_1^{m,s}, \mathbf{z}_2^{m,s}, ..., \mathbf{z}_N^{m,s}] \in \mathbb{R}^{N \times D_m}$. Inspired by [16], we employ a decoder $\mathcal{D}^m$ to reconstruct $\mathbf{z}_i^{m,s}$, which ensures learning sufficient features from the raw data for clustering those relevant information, i.e.,

$$\hat{\mathbf{x}}_i^{m,s} = \mathcal{D}^m \left( \mathbf{z}_i^{m,s} \right) = \mathcal{D}^m \left( \mathcal{E}^m \left( \mathbf{x}_i^{m,s} \right) \right). \tag{2}$$

The reconstruction loss involving all modalities is defined as:

$$\begin{aligned} \mathcal{L}_{\text{intra}}^{\text{rec}} &= \frac{1}{MS_m} \sum_{m=1}^M \sum_{s=1}^{S_m} \left\| \mathbf{X}^{m,s} - \hat{\mathbf{X}}^{m,s} \right\|_2^2 \\ &= \frac{1}{MS_m N} \sum_{m=1}^M \sum_{s=1}^{S_m} \sum_{i=1}^N \left\| \mathbf{x}_i^{m,s} - \mathcal{D}^m \left( \mathcal{E}^m \left( \mathbf{x}_i^{m,s} \right) \right) \right\|_2^2. \end{aligned} \tag{3}$$

To extract discriminative features from data of each modality, we use an adaptive weighting technique to obtain the intra-modal mean feature $\bar{Z}^m \in \mathbb{R}^{N \times D_m}$, where $\bar{Z}^m = [\bar{\mathbf{z}}_1^m, \bar{\mathbf{z}}_2^m, \cdots, \bar{\mathbf{z}}_N^m]$ and $\bar{\mathbf{z}}_i^m$ is defined as:

$$\bar{\mathbf{z}}_i^m = \sum_{s=1}^{S_m} w_{m,s} \mathbf{z}_i^{m,s}. \tag{4}$$

Here, $w_{m,s}$ denotes the $s$-th item's weight of the $m$-th modality and $\sum_{s=1}^{S_m} w_{m,s} = 1$. The weights are optimized during training, reflecting the importance of different items within a single modality.

To optimize the above weights, we adopt a two-layer linear MLP $\mathcal{F}_{\text{intra}}^m$ to map the modal-specific features and the mean intra-modal feature to a $K$-dimension space. Here, $K$ is the number of clusters. Then, a softmax layer for normalization is attached to obtain the probability of each feature belonging to each cluster. That is, we obtain the probability matrix for each intra-modal item of each subject, i.e., $C^{m,s} = [c_1^{m,s}, c_2^{m,s}, \cdots, c_N^{m,s}] \in \mathbb{R}^{N \times K}$ and that for the intra-modal mean features as well, i.e., $\bar{C}_{\text{intra}}^m = [\bar{c}_1^m, \bar{c}_2^m, \cdots, \bar{c}_N^m] \in \mathbb{R}^{N \times K}$. Specifically, $c_{i,j}^{m,s}$ denotes the probability of the $s$-th item of the $i$-th subject from the $m$-th modality that belongs to the cluster $j$. Since the intra-modal items from the same cluster should have similar features, cluster contrastive learning decreases the distance between item pairs from the same cluster while increasing the distance between those pairs from distinct clusters. Therefore, $\{\bar{c}_{i,j}^m, c_{i,j}^{m,s}\}_{s=1,2,\cdots,S_m}$ are positive pairs from subject $i$, while other pairs of $\{\bar{c}_{i,j}^m, c_{i,\neq j}^{m,s}\}_{s=1,2,\cdots,S_m}$ are negative pairs.

To quantify the similarity between two clusters, we use the Cosine similarity as follows:

$$S \left( \bar{c}_{i,j}^m, c_{i,j}^{m,s} \right) = \frac{\bar{c}_{i,j}^m \cdot c_{i,j}^{m,s}}{\left\| \bar{c}_{i,j}^m \right\| \left\| \mathbf{c}_{i,j}^{m,s} \right\|}. \tag{5}$$

Based on this metric, we define the intra-modal cluster contrastive loss as

$$\mathcal{L}_{\text{intra}}^{\text{con}} = -\frac{1}{NKMS_m} \sum_{i=1}^N \sum_{j=1}^K \sum_{m=1}^M \sum_{s=1}^{S_m} \log \frac{e^{S \left( \bar{c}_{i,j}^m, c_{i,j}^{m,s} \right)/\tau_1}}{\sum_{k=1}^K e^{S \left( \bar{c}_{i,j}^m, c_{i,k}^{m,s} \right)/\tau_1} - e^{1/\tau_1}}. \tag{6}$$

Here, $\tau_1$ denotes the temperature coefficient. The total intra-modal loss is:

$$\mathcal{L}_{\text{intra}} = \mathcal{L}_{\text{intra}}^{\text{res}} + \mathcal{L}_{\text{intra}}^{\text{con}}. \tag{7}$$

*3.2.2 Inter-modal Contrastive Learning.* Similarly, we apply inter-modal contrastive learning to learn modality consistency across multiple modalities. We also use the above adaptive weighting technique to obtain the inter-modal mean features $\bar{\bar{\mathbf{Z}}} = [\bar{\bar{\mathbf{z}}}_1, \bar{\bar{\mathbf{z}}}_2, \cdots, \bar{\bar{\mathbf{z}}}_N] \in \mathbb{R}_{N \times D}$, where each $\bar{\bar{\mathbf{z}}}_i$ is computed as:

$$\bar{\bar{\mathbf{z}}}_i = \sum_{m=1}^M w_m \bar{\mathbf{z}}_i^m. \tag{8}$$

Here, $w_m$ is the weight of the $m$-th modality and $\sum_{m=1}^M w_m = 1$. To optimize these weights, we adopt another two-layer MLP $\mathcal{F}_{\text{inter}}$ and another Softmax layer to obtain the clustering probability, i.e., $\hat{\bar{C}}_{\text{inter}}^m = [\hat{c}_1^m, \hat{c}_2^m, \cdots, \hat{c}_N^m] \in \mathbb{R}^{N \times K}$ for each modality feature $\bar{\mathbf{z}}_i^m$ and

$\bar{\bar{C}}_{inter} = [\bar{\bar{c}}_1, \bar{\bar{c}}_2, \cdots, \bar{\bar{c}}_N] \in \mathbb{R}^{N \times K}$ for the inter-modal mean feature $\bar{\bar{z}}_i$. Similarly, $\hat{\bar{c}}_{i,j}^m$ represents the probability of the $m$-th inter-modal feature of $i$-th subject that belongs to cluster $j$. For inter-modal mean feature $\bar{\bar{c}}_{i,j}$, $\{\bar{\bar{c}}_{i,j}, \hat{\bar{c}}_{i,j}^m\}_{m=1,2,\cdots,M}$ are positive pairs and the rest cluster pairs of $\{\bar{\bar{c}}_{i,j}, \hat{\bar{c}}_{i,\neq j}^m\}_{m=1,2,\cdots,M}$ are negative pairs. Therefore, the inter-modal contrastive loss is:

$$\mathcal{L}_{inter} = -\frac{1}{NKM} \sum_{i=1}^{N} \sum_{j=1}^{K} \sum_{m=1}^{M} \log \frac{e^{S(\bar{\bar{c}}_{i,j}, \hat{\bar{c}}_{i,j}^m)/\tau_1}}{\sum_{k=1}^{K} e^{S(\bar{\bar{c}}_{i,j}, \hat{\bar{c}}_{i,k}^m)/\tau_1} - e^{1/\tau_1}}. \quad (9)$$

## 3.3 Multimodal Relevance-gated Attention Transformer (MRAT)

Transformers [53] have demonstrated their superior advantages in modeling different modalities, e.g., visual, language, and audio, on various multimodal tasks, e.g., visual question answering [1, 30, 39], vision-language pre-training [4, 5, 55]. Therefore, we adopt the transformer architecture to fuse our multimodal weighted features, as shown in Fig 2(b). After obtaining image and text features and their relevance scores from our SMRW module, we fed them into a feature enhancer for cross-modality feature fusion. The feature enhancer includes two feature enhancer layers, i.e., an image-to-text cross-attention and a text-to-image cross-attention for feature fusion. These modules help align features of different modalities, which are fed into a feed-forward network (FFN) layer, respectively. After such deep interaction between image and text features, we concatenate and pass them into a self-attention layer for modality alignment and fusion and finally output the diagnostic prediction.

Different from the commonly-used self- and cross-attention layers, we propose the relevance-gated attention operations to fully leverage the relevance scores learned during the SMRW module. Specifically, our attention block is defined as:

$$\begin{aligned} &\textbf{AttentionBlock}(Q, K, V, R) \\ &= \text{LN}(Q + \text{FFN}(Q + \text{Attn}(Q, K, V, R))), \end{aligned} \quad (10)$$

where $Q, K, V$ represents the query, key, and value features, $R$ is the relevance score obtained from SMRW, $\text{LN}(\cdot)$ represents a layer normalization, and the feed-forward network, $\text{FFN}(\cdot)$, is a sequence of layer normalization, linear transformation, GELU activation, and another linear transformation. Lastly, the attention mechanism, $\text{Attn}(Q, K, V, R)$, is calculated as follows:

$$\begin{aligned} Q' &= \text{Linear}(Q), & R' &= \text{LN}(\text{Linear}(R)), \\ K' &= \text{Linear}(K \oplus R'), & V' &= \text{Linear}(V), \\ \text{Attn}(Q, K, V, R) &= \text{Softmax}\left(Q'K'^{\top}\right) V'. \end{aligned} \quad (11)$$

Here, $\oplus$ denotes addition along the feature dimension, and $\text{Linear}(\cdot)$ represents a linear transformation.

## 3.4 Loss Function

We employ three loss functions to train our model, including one classification loss and two self-supervised relevance weighting losses, i.e., the intra- and inter-modal contrastive learning losses.

**Table 1: The statistics of the datasets in our experiments.**

| Datasets | Type | #Image | #Non-imaging clinical data | | | | |
|---|---|---|---|---|---|---|---|
| | | | SC | MH | NS | BA | GN |
| ADNI [42] | NC/MCI/AD | 1042 | 16 | 20 | 31 | 34 | 12 |
| AIBL [13] | NC/MCI/AD | 858 | 2 | 18 | 4 | 12 | 4 |
| | | | SC | BS | MH | M | NM |
| PPMI [37] | NC/PD | 599 | 11 | 24 | 12 | 34 | 36 |

SC: Subject characteristics, MH: Medical History, NS: Neuropsychology Scores, BA: Blood Analysis, GN: Genetics, BS: BIO-Specimen, M: Motor Function, NM: Non-Motor Function.

In the training process, the classification loss is constructed by a cross-entropy function:

$$\mathcal{L}_{cls} = -\frac{1}{NC} \sum_{i=1}^{N} \sum_{c=1}^{C} y_i^c \log\left(p_i^c\right), \quad (12)$$

where $p_i^c$ is the predicted classification score of each subject for each class $c$, while $y_i^c$ is the corresponding one-hot ground-truth label. The overall loss is defined below:

$$\mathcal{L} = \mathcal{L}_{cls} + \alpha \cdot \mathcal{L}_{intra} + \beta \cdot \mathcal{L}_{inter}, \quad (13)$$

where $\mathcal{L}_{intra}$ and $\mathcal{L}_{inter}$ represent the intra-modal contrastive loss given in Eq. (7) and the inter-modal one in Eq. (9), respectively; $\alpha$ and $\beta$ are hyper-parameters balancing these three terms.

## 4 EXPERIMENTS

### 4.1 Datasets

Three public multimodal datasets are evaluated in this study, with considering both medical image scans (e.g., structure MRIs) and non-imaging clinical information (e.g., demographics, serological tests, neuropsychology examinations). The detailed statistics of these datasets are summarized in Table 1.

*ADNI [42].* The ADNI dataset is a longitudinal and multi-site multimodal neuroimaging dataset. We collect 1042 subjects for evaluation, including 342 normal controls (NC), 351 with mild cognitive impairment (MCI), and 349 patients with Alzheimer's disease (AD) from ADNI1, ADNI2, ADNIGO, and ADNI3. We excluded follow-up scans and included a single baseline scan of the structure MRI and non-imaging clinical information per subject. These subjects are divided into three groups (i.e., AD, MCI, and NC), and MCI is considered to be a significant stage for the preclinical AD diagnosis.

*AIBL [13].* A total of 858 subjects are included in this dataset, where 609 NCs, 144 MCIs, and 105 AD subjects were recruited from the same technical infrastructure as the ADNI. AIBL is also a longitudinal dataset and we process it in the same way with ADNI.

*PPMI [37].* The PPMI dataset includes 247 NCs and 352 PDs. The patients were diagnosed at baseline, and the NCs were healthy at their first examination. Notably, the PPMI dataset is also a longitudinal database, and each participant has multiple scans. We also included

**Table 2: Comparison among different methods on AD classification using ADNI and AIBL datasets. The best results are in bold.**

| Methods | Modality | ADNI | | | | | | | | AIBL | | | | | | | |
|---|---|---|---|---|---|---|---|---|---|---|---|---|---|---|---|---|---|
| | | NC vs MCI | | MCI vs AD | | NC vs AD | | NC vs MCI vs AD | | NC vs MCI | | MCI vs AD | | NC vs AD | | NC vs MCI vs AD | |
| | | *ACC* | *AUC* | *ACC* | *AUC* | *ACC* | *AUC* | *ACC* | *AUC* | *ACC* | *AUC* | *ACC* | *AUC* | *ACC* | *AUC* | *ACC* | *AUC* |
| Res50 [20] | I. | 72.77 | 79.13 | 73.14 | 80.43 | 77.25 | 82.14 | 65.22 | 73.91 | 74.61 | 81.08 | 75.48 | 81.60 | 78.73 | 82.14 | 68.29 | 75.15 |
| Med3D [6] | I. | 75.25 | 81.40 | 75.77 | 81.34 | 80.06 | 83.75 | 69.57 | 72.64 | 76.90 | 79.78 | 77.14 | 81.13 | 81.30 | 85.25 | 72.73 | 74.91 |
| ViT [11] | I. | 83.60 | 86.93 | 84.65 | 88.16 | 88.67 | 90.95 | 79.83 | 84.73 | 84.48 | 88.26 | 85.61 | 88.89 | 90.46 | 91.06 | 76.19 | 81.40 |
| M3T [23] | I. | 86.14 | 89.17 | 86.06 | 89.73 | 89.64 | 91.06 | 74.77 | 78.51 | 86.95 | 89.73 | 87.16 | 91.14 | 90.02 | 92.27 | 75.47 | 81.55 |
| BERT [10] | T. | 88.21 | 90.04 | 88.91 | 91.23 | 92.04 | 93.45 | 80.80 | 83.57 | 88.46 | 90.38 | 91.70 | 93.73 | 91.27 | 93.34 | 81.14 | 84.82 |
| RoBERTa [36] | T. | 88.80 | 90.89 | 89.17 | 92.47 | 92.12 | 93.73 | 81.36 | 84.75 | 88.18 | 91.83 | 90.73 | 93.77 | 92.87 | 94.26 | 83.27 | 85.93 |
| Perceiver [22] | I.+T. | 86.48 | 89.75 | 87.12 | 89.75 | 90.55 | 92.34 | 73.91 | 77.85 | 87.24 | 89.83 | 88.29 | 90.67 | 91.06 | 92.39 | 72.73 | 77.39 |
| GIT [54] | I.+T. | 85.36 | 87.59 | 86.11 | 89.76 | 89.17 | 91.75 | 82.46 | 83.87 | 86.22 | 86.36 | 88.02 | 89.34 | 91.17 | 91.56 | 80.67 | 82.86 |
| Irene [64] | I.+T. | 88.65 | 91.94 | 89.74 | 90.45 | 91.71 | 93.54 | 83.48 | 85.34 | 88.32 | 89.64 | 91.30 | 93.53 | 93.36 | 94.56 | 84.27 | 86.25 |
| Alifuse [5] | I.+T. | 88.21 | 90.93 | 89.73 | 90.59 | 91.57 | 93.54 | 85.93 | 87.36 | 89.65 | 91.94 | 89.29 | 91.86 | 93.48 | **95.76** | 86.31 | 88.99 |
| SMART (ours) | I.+T. | **90.17** | **92.16** | **91.06** | **92.73** | **93.56** | **94.82** | **89.18** | **90.21** | **91.21** | **93.25** | **92.73** | **94.54** | **93.81** | 94.63 | **89.92** | **91.45** |

**Table 3: Comparison of different approaches on PD classification using the PPMI dataset. The best results are in bold.**

| Methods | Modality | NC vs PD | |
|---|---|---|---|
| | | *ACC* | *AUC* |
| Res50 [20] | I. | 73.12 | 76.72 |
| Med3D [6] | I. | 76.45 | 78.84 |
| ViT [11] | I. | 75.45 | 79.56 |
| M3T [23] | I. | 75.84 | 78.21 |
| BERT [10] | T. | 79.24 | 81.84 |
| RoBERTa [36] | T. | 80.53 | 81.46 |
| Perceiver [22] | I.+T. | 77.64 | 80.35 |
| GIT [54] | I.+T. | 83.57 | 85.53 |
| Irene [64] | I.+T. | 84.43 | 87.75 |
| Alifuse [5] | I.+T. | 85.25 | 88.63 |
| SMART(ours) | I.+T. | **87.12** | **89.34** |

only the baseline scan of the structure MRI and non-imaging clinical information per subject.

## 4.2 Data preprocessing

As shown in Table 1, we collect over a hundred pieces of non-imaging clinical data for each patient in the ADNI dataset. For instance, there are 20 medical histories for each patient. Clinical text data are semantically prepared before encoding for all datasets. Clinical variable terms are collected and cleaned using the ADNI Data Dictionary or PPMI Data Dictionary, and then semantically aggregated to the appropriate level of granularity to ensure a less sparse dataset. For example, if a patient's HMT3=4.39, a corresponding sentence is written as "Red blood cell count is 4.39". If a patient's MH4CARD=1, then it is described as "has a cardiovascular medical history". We argue that its contained digital numbers and categories

are more meaningful in the text context. Therefore, we create a text representation for each data sample using the information contained in the tables of raw data. Regarding the collected T1-weighted structural MRI (sMRI) scans, we apply the same data pre-processing to normalize and standardize them from a multi-institutional database. We resize the images to have the same voxel spacing (i.e., 1.75mm × 1.75mm × 1.75mm) and the same volume size (i.e., 128 × 128 × 128), then normalize the image intensities of all the voxels using the zero-mean unit-variance method.

## 4.3 Implementation details

The experimental framework is implemented on the Pytorch platform and executed on a 24GB NVIDIA Geforce RTX 3090 Linux server. The image patch size is set to 16 × 16 × 16. The encoder and decoder in SMRW are implemented using fully connected layers with the encoder dimensions of input-500-1000-1000-512. The decoder is symmetric with the encoder. We set hyper-parameters $\alpha$ and $beta$ both to 1. Adam optimizer is adopted with a learning rate of 3e-4 and a batch size of 32, trained for 300 epochs. For all experiments, we evaluated the performance in terms of two metrics, i.e., accuracy (ACC) and area under curve (AUC).

## 4.4 Experimental Results

*Baselines.* In this study, we compare our proposed SMART with baseline deep learning approaches and well-estimated transformer-based methods. These methods include: (1) the **image-only** group. We choose four recent baselines that are commonly used, including Net50 [20], Med3D [6], VisionTransformer [11], M3T [23]. (2) The **text-only** group. We choose BERT [10] and RoBERTa [36] as our text-only baseline. (3) the **multi-modal** group. We have four baselines, i.e., GIT [54], Preciver [22], IRENE [64] and Alifuse [5]. They are recent transformer-based models that fuse multi-modal information for classification. IRENE and Alifuse are previous SOTA methods designed for multimodal medical diagnosis.

*Comparison Results.* Table 2 and Table 3 report the comparison results of our SMART with ten baseline methods on three datasets

**Table 4: Ablation studies on each component of our model by classifying NC, MCI, and AD subjects.**

| Method | ADNI | | AIBL | |
|---|---|---|---|---|
| | *ACC* | *AUC* | *ACC* | *AUC* |
| (a) SMART w/o $\mathcal{L}_{intra}^{res}$ | 88.05 | 88.37 | 86.56 | 87.21 |
| (b) SMART w/o $\mathcal{L}_{intra}$ | 82.84 | 83.42 | 82.55 | 84.02 |
| (c) SMART w/o $\mathcal{L}_{inter}$ | 88.73 | 89.06 | 89.17 | 90.46 |
| (d) SMART w/o SMRW | 73.67 | 76.51 | 77.24 | 80.17 |
| (e) SMART w/o MRAT | 85.32 | 87.10 | 86.31 | 87.25 |
| (f) SMART-I | 76.52 | 77.97 | 77.89 | 79.34 |
| (g) SMART-T | 87.67 | 89.24 | 87.56 | 86.93 |
| (h) SMART-S | **89.18** | **90.21** | **89.92** | **91.45** |

for classifying two representative neurodegenerative diseases, i.e., AD and PD. As shown in Table 2, on all datasets SMART significantly outperforms the image-only model, the text-only model, and four recent SOTA transformer-based multimodal methods, except for one AUC result provided by Alifuse. Take three-label classification on ADNI for example, SMART achieves the highest accuracy of 0.89, over ~15% higher than the image-only model that only takes structural MRIs as input. In comparison with the text-only method, SMART maintains an advantage of over ~8% improvement. Comparing SMART to GIT, we observe an advantage of over ~7%. Compared to previous SOTA methods, i.e., Irene and Alifuse, the transformer-based multimodal classification model, SMART still delivers competitive results, surpassing them by ~4%.

## 4.5 Ablation Studies

We conduct ablation studies on ADNI and AIBL to evaluate the crucial factors in our proposed SMART, as shown in Table 4. Here we provide some implementation details. (a) "w/o $\mathcal{L}_{intra}^{res}$": we remove our intra-modal reconstruction loss in the SMRW module in the training process. (b) "w/o $\mathcal{L}_{intra}$": we remove our intra-modal constrastive learning loss in the training process. (c) "w/o $\mathcal{L}_{inter}$": we remove our inter-modal constrastive learning loss in the training process. (d) "w/o SMRW": we remove the SMRW module and replace the task-relevance score with a randomly initialized learnable tensor, which has the same shape as the value features $V$. (e) "w/o MRAT": we remove the guidance of relevance in MRAT by replacing the task-relevance score with a zero tensor, which has the same shape as $V$ as well. (f) "-I": we only train the visual modality in SMRW and remove its textual branch by replacing the textual feature with zero tensor. (g) "-T": we perform a similar operation as (f), except that we use the prediction of the textual branch during inference. (h) "-S": our standard model with both images and text.

*Importance of SMRW.* Checking the d and h rows in Table 4, we observe that the prediction accuracy improves significantly when trained with self-supervised multimodal relevance weighting, verifying the effectiveness of our self-weighted task relevance technique. Taking one step further, on the ADNI dataset we achieve ~15% performance improvement on ACC by using SMRW, and ~14% performance improvement on AUC; while on AIBL, these

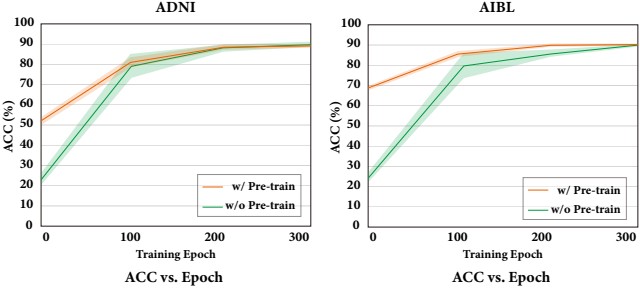

**Figure 4: Ablation on different pretraining strategy.**

two numbers are about 14%, 11%. This indicates that our SMRW module can alleviate the information irrelevancy efficiently especially when facing a large amount of data. We further perform ablation studies on loss functions in SMRW, which are reported in rows of (a,b,c), intra-modal contrastive learning is more helpful than inter-modal contrastive learning. Also, it suggests that to prevent the model from crashing, reconstruction regularization is crucial on smaller datasets with varying data quality.

*Effectiveness of MRAT.* Comparing the row (e) with (h) in Table 4, there are great performance drops of ~4% on ACC of ADNI and 3% on ACC of AIBL, if the relevance-gated attention is removed. This indicates that the guidance of the relevance score is essential for a more accurate diagnosis and reduces the difficulty of fusing high-dimensional multimodal data.

*Benefits of Multimodal Learning.* Comparing the rows (f,g) with (h) in Table 4, there are also dramatic drops of 11%, 12% on ACC and AUC of ADNI if the text modality is removed, and 1.5%, 2% drops on ACC and AUC when the image modality is removed. This ablation study demonstrates that the multi-modal design outperforms the unimodal one, showing the necessity of using both imaging and non-imaging clinical data for neurodegenerative disorder diagnosis. Compared to the imaging data, the non-imaging data is more effective in classifying AD. Thus, it is worth to discover and amplify the crucial clues within the non-imaging clinical text data.

*To Pretrain or Not?* Since our SMRW module is self-supervised, one potential attempt would be using the self-supervised loss in SMRW to pre-train. For instance, we first pre-train the SMRW module for 100 epochs, which is followed by another 200 epochs using the overall loss functions. As shown in Figure 4, the earlier stage performance improvement is stable with pre-training SMRW. Interestingly, in the ADNI experiment, the later adjustment in the optimization process weakens the model's performance. This indicates that suitable pretraining is a good choice while excessive adjustment using supervised signals would discard some valid information learned in the early stage of training.

## 4.6 Qualitative Analysis

Figure 5 shows the visual comparison between SMART and the baseline, i.e., our method witout the SMRW module. The image patches and text sentences with the top five highest scores are selected for comparison. We observe that the baseline model misses

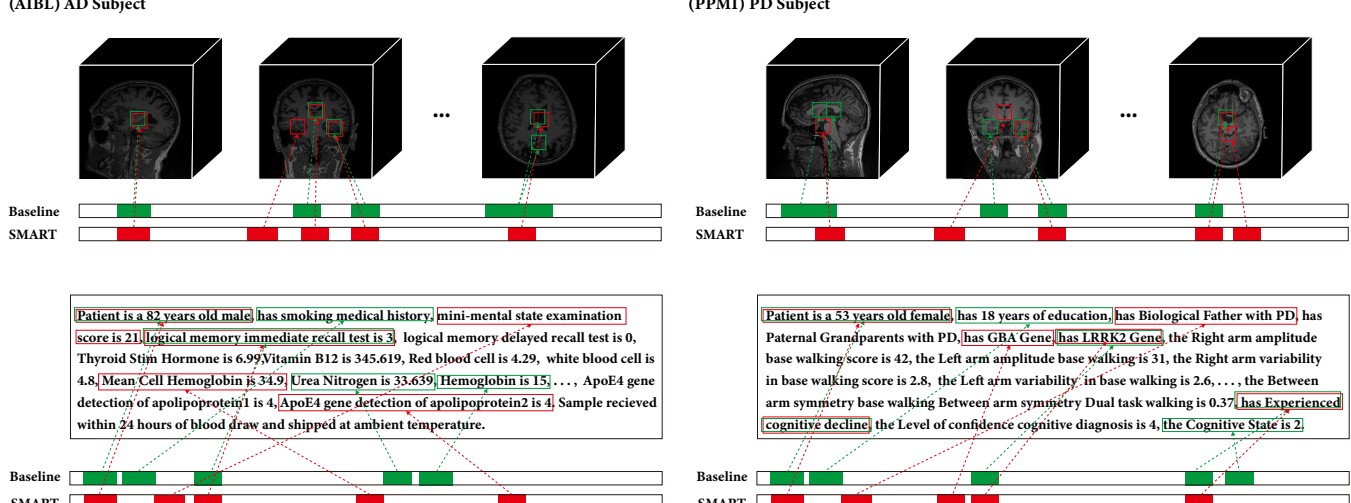

**Figure 5: Visualization of relevance scores on sub-regions of brain MRI scans and sentences of non-imaging text description. "Baseline" indicates our SMART without the SMRW module.**

Patient is a 82 years old male, has smoking medical history, mini-mental state examination score is 21, logical memory immediate recall test is 3, logical memory delayed recall test is 0, Thyroid Stim Hormone is 6.99,Vitamin B12 is 345.619, Red blood cell is 4.29, white blood cell is 4.8, Mean Cell Hemoglobin is 34.9, Urea Nitrogen is 33.639, Hemoglobin is 15, . . . , ApoE4 gene detection of apolipoprotein1 is 4, ApoE4 gene detection of apolipoprotein2 is 4. Sample recieved within 24 hours of blood draw and shipped at ambient temperature.

**Alifuse**

Patient is a 82 years old male, has smoking medical history, mini-mental state examination score is 21, logical memory immediate recall test is 3, logical memory delayed recall test is 0, Thyroid Stim Hormone is 6.99,Vitamin B12 is 345.619, Red blood cell is 4.29, white blood cell is 4.8, Mean Cell Hemoglobin is 34.9, Urea Nitrogen is 33.639, Hemoglobin is 15, . . . , ApoE4 gene detection of apolipoprotein1 is 4, ApoE4 gene detection of apolipoprotein2 is 4. Sample recieved within 24 hours of blood draw and shipped at ambient temperature.

**SMART**

**Figure 6: Visualization of the attention maps on the non-imaging clinical data of an AD subject sampled from the AIBL dataset. A darker blue indicates a higher attention.**

to pay great attention on some important biomarkers[25, 29, 46]. On the contrary, our model demonstrates better locations of those key image subregions and text sentences.

To further demonstrate the interpretability of our model, we compare SMART with the previous SOTA method Alifuse. Figure 6 shows the attention weight distribution over the text description of the non-imaging clinical data. We discover that our model can capture the feature of a longer sequence. In the sentence "ApoE4 gene detection of apolipoprotein", which is a biomarker of Alzheimer's disease [29], Alifuse exerts little attention, while our model does capture task relevance between the meaning of the words and the neurodegenerative disease. Our method demonstrates its ability to handle a long text description to some extent. This is essential in

the diagnostics of neurodegenerative disorders since a lot of non-imaging clinical information is desired to be explored to identify the unknown yet potential biomarkers for diagnosis.

## 5 CONCLUSION

In this paper, we propose a novel diagnosis framework SMART that fuses multimodal medical data via a self-supervised weighting approach and augments transformer attention techniques to achieve a more accurate diagnostic classification. We demonstrate the effectiveness of our model in classifying both AD and PD, which outperforms ten baselines and achieves the SOTA classification accuracy. Currently, we only consider brain MRI scans; in future work, more image modalities like PET and even others like audio will be included as well for more accurate diagnosis. Another extension to our current model is the incorporation of longitudinal data. Because degeneration over age is an essential characteristic of neurodegenerative disorders, multimodal fusion on longitudinal data will be our future work as well.

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
