# OpenReview forum: "SMART: Self-Weighted Multimodal Fusion for Diagnostics of Neurodegenerative Disorders"
_acmmm.org/ACMMM/2024/Conference — MM2024 Poster_

### Official Review · Reviewer_SrqY · 2024-05-15

**Rating:** 4
**Confidence:** 3

**Summary:**

The paper introduces a novel approach called SMART that leverages a deep self-weighted multimodal relevance weighting technique to eliminate irrelevant information and improve fusion for accurate diagnosis. By integrating clustering-based contrastive learning and a multimodal attention transformer, the proposed model achieves state-of-the-art performance on three public datasets. The key contributions include the development of SMART, the introduction of the self-weighted multimodal representation learning technique SMRW, and the explainability and theoretical design for multiple modalities.

**Strengths:**

1. The logic of this work is straightforward.
2. The motivation behind this work is clear.
3. The diagram of the framework is helpful to understand.
4. The methodology is reasonable.
5. The experimental section is comprehensive.

**Limitations:**

1. This work claims that the proposed method allows efficient multimodal feature fusion; however, it does not provide evidence to support its efficiency. Why is the proposed method considered efficient? Could you please explain and provide supporting evidence?

2. The proposed framework lacks novelty to some extent. It is a combination of well-explored techniques.

3. There is no source code provided, and the experiment results show that the proposed method almost outperforms all baseline methods in all cases, which is infrequent. I kindly request that you check the experiment setting to avoid unexpected mistakes like data leakage.

3. Some typo mistakes, e.g, 4.4 Experimental results, (3)
"the multi-modal ..." should be "The multi-modal..."

**Suitability:**

3

---

### Official Review · Reviewer_PLwt · 2024-05-26

**Rating:** 4
**Confidence:** 4

**Summary:**

The paper introduces SMART, a framework for diagnosing neurodegenerative disorders, outperforming ten baselines across three benchmark datasets. Key contributions include a novel self-weighted multimodal representation learning technique, SMRW, which enhances model explainability while maintaining high diagnostic accuracy. SMART is designed to incorporate multiple modalities, potentially improving diagnostic capabilities by leveraging diverse medical information sources.

**Strengths:**

One strength of this paper lies in its introduction of a SMART framework, which demonstrates superior performance over existing methods across multiple benchmark datasets. Additionally, the proposed self-weighted multimodal representation learning technique, SMRW, contributes to the model's interpretability while maintaining high diagnostic accuracy. Furthermore, the model's theoretical design allows for the incorporation of multiple modalities, potentially enhancing its diagnostic capabilities by leveraging a wide range of medical information sources.

**Limitations:**

1. Why authors did not use more advanced metrics like precision, recall, specificity, f1 score, and so on included with AUC and accuracy metrics to evaluate their proposed approach performance?

2. Why authors did not use PSNR and SSIM metrics to compare reconstructed images with original images? In addition, this paper does not have a qualitative analysis regarding the reconstructed images which can able to compared with the original images.

3. the authors should discuss the impact of each modality employed in the study on enhancing robust performance in the results section.

4. The authors should conduct thorough performance comparisons with recent existing multimodal approaches to provide insights into the effectiveness of their proposed method. It would enhance the paper's contribution and provide valuable context for readers.

5. A detailed discussion is required because how complementary information from different modalities influenced the improvement of the learning model's performance is essential. Exploring the synergies between modalities and analyzing how they contribute to performance enhancement would provide valuable insights for researchers and practitioners in the field.

6. Regarding the writing of the method section lacks proper presentation according to these highly reputed conference standards, as there is no clear problem formulation provided to enhance the understanding of the paper better.

**Suitability:**

3

---

### Official Review · Reviewer_bE32 · 2024-05-30

**Rating:** 4
**Confidence:** 4

**Summary:**

The paper discusses the challenges and solutions in diagnosing neurodegenerative disorders, such as Alzheimer's disease (AD) and Parkinson's disease (PD), using multimodal medical data. This data includes brain scans and non-imaging clinical records like demographics and neuropsychology examinations. The main issue in fusing these multimodal inputs lies in the overwhelming amount of irrelevant information contained within the high-dimensional data.

**Strengths:**

1. Novelty and Theoretical Approach
The paper introduces a novel method called SMART (Self-weighted Multimodal Attention-and-Relevance gated Transformer), which leverages deep self-weighted multimodal relevance weighting combined with clustering-based contrastive learning. This innovative approach effectively addresses the challenge of representation degeneration in high-dimensional medical data by filtering out irrelevant information, ensuring a more accurate and robust feature fusion process.

2. Technical Correctness and Adequate Evaluation
The proposed methodology is technically sound, with a clear and detailed description of the algorithms and model architecture. The SMART model is evaluated extensively on three public datasets for Alzheimer's disease (AD) and Parkinson's disease (PD), where it achieves state-of-the-art (SOTA) performance. Comparisons with existing methods provide strong evidence of its superior accuracy and robustness.

3. Clarity and Practical Applications
The paper is well-structured and clearly written, making complex concepts accessible. The practical implications are significant, particularly in improving the diagnosis of neurodegenerative disorders. By enhancing diagnostic accuracy, SMART has the potential to positively impact clinical decision-making and patient outcomes. Furthermore, the authors' commitment to sharing their source code promotes transparency and encourages further research in this domain.

**Limitations:**

Weakness: Lack of Discussion on Related Work <<Visual-Attribute Prompt Learning for Progressive Mild Cognitive Impairment Prediction>> in MICCAI 2023
1. Insufficient Literature Context
The paper does not adequately discuss related work in the area of visual-attribute prompt learning, particularly as it applies to progressive mild cognitive impairment (MCI) prediction. This omission leaves a gap in the understanding of how the proposed SMART model compares or relates to recent advancements in this specific domain. For instance, visual-attribute prompt learning has shown promise in dynamically integrating visual features with clinical attributes, which could provide valuable insights or complementary techniques for improving the diagnosis of neurodegenerative disorders.

By not including a discussion on visual-attribute prompt learning approaches, the authors miss an opportunity to compare and contrast their method with potentially relevant and recent methodologies. This comparison is crucial as it can highlight the unique contributions of the SMART model and provide a clearer understanding of its relative strengths and weaknesses. For example, examining how visual-attribute prompt learning handles feature fusion and relevance weighting could offer insights into possible enhancements or alternative strategies for the SMART framework.

**Suitability:**

2

---

### Official Review · Reviewer_b21z · 2024-06-04

**Rating:** 3
**Confidence:** 3

**Summary:**

This work introduces SMART, a deep self-weighted multimodal correlation weighting method that utilizes cluster-based constraint learning to eliminate intra- and intermodal uncorrelations. The learned correlation scores are integrated as gates with multimodal attention converters to provide improved fusion for the final diagnosis and validated on three datasets (AD/PD).

**Strengths:**

1.The multimodal problem is widespread in the field of multimodel medical data classification. However, previous work in this field has not dealt with it properly. This work provides an excellent formulation of the problem for describing this study and proposes valuable solutions that can be used to inform subsequent research.
2.The paper is very well organized. The research question is fully described.
3.Validation was carried out on three datasets, which is important to demonstrate its generalization.

**Limitations:**

1. One weakness of the paper is that some architectures, experimental designs, and parameter choices are not explained and not proven. For example, the choice of cosine as the similarity function, the effect of two-layer MLP on intra-modal mean feature, etc
2. Again, while listing the hyperparameters used in the model can help improve repeatability, this alone is not enough because their choices are irrational. A search in the main paper shows no mention of hyperparameter tuning.
3.In Table 1, the AIBL dataset has the least clinical data with class-imbalance, but best results are achieved in Tables 2 and 3, and the reasons for this phenomenon need to be explained.
4.Other missing but required elements include: How is the entire framework trained? In particular, self-supervised modules and the whole networks. We know that clinical data is often insufficient, what is the training strategy for this work?
5. In the experiment, the results of the mean ± standard deviation are more convincing.

**Suitability:**

3

---

### Meta-Review · Area_Chair_Rowh · 2024-06-28

**Recommendation:** Accept (Poster)
**Confidence:** 5

**Metareview:**

The paper introduces SMART, a deep self-weighted multimodal correlation weighting method that uses cluster-based constraint learning to eliminate intra- and intermodal uncorrelations. This method is designed to improve the fusion of multimodal data for diagnosing neurodegenerative disorders, specifically Alzheimer’s disease (AD) and Parkinson’s disease (PD). The proposed approach is validated on three datasets, demonstrating its effectiveness and generalization capabilities.
The paper presents a well-structured and technically sound approach to multimodal medical data classification, addressing a significant challenge in diagnosing neurodegenerative disorders. Despite some areas needing further explanation and justification, the method's validation on multiple datasets and the comprehensive experiments provide strong evidence of its effectiveness. The paper's contributions are valuable to the field and warrant acceptance as a poster.